# Forest Type and Climate Outweigh Soil Bank in Shaping Dynamic Changes in Macrofungal Diversity in the Ancient Tree Park of Northeast China

**DOI:** 10.3390/jof9080856

**Published:** 2023-08-17

**Authors:** Zhaoxiang Zhu, Xin Liu, Tom Hsiang, Ruiqing Ji, Shuyan Liu

**Affiliations:** 1Engineering Research Center of Edible and Medicinal Fungi, Ministry of Education, Jilin Agricultural University, Changchun 130118, China; zhuzx1985@163.com (Z.Z.); codelj741@163.com (X.L.); 2Department of Environmental Sciences, University of Guelph, Guelph, ON N1G 2W1, Canada; thsiang@uoguelph.ca

**Keywords:** macrofungal diversity, forestry types, soil fungal bank, climate, Changbai Mountain

## Abstract

The community structure of macrofungi is influenced by multiple complex factors, including climate, soil, vegetation, and human activities, making it challenging to discern their individual contributions. To investigate the dynamic changes in macrofungal diversity in an Ancient Tree Park located in Northeast China and explore the factors influencing this change, we collected 1007 macrofungi specimens from different habitats within the park and identified 210 distinct fungal species using morphological characteristics and ITS sequencing. The species were classified into 2 phyla, 6 classes, 18 orders, 55 families, and 94 genera. We found macrofungal compositions among different forest types, with the mixed forest displaying the highest richness and diversity. Climatic factors, particularly rainfall and temperature, positively influenced macrofungal species richness and abundance. Additionally, by analyzing the soil fungal community structure and comparing aboveground macrofungi with soil fungi in this small-scale survey, we found that the soil fungal bank is not the main factor leading to changes in the macrofungal community structure, as compared to the influence of climate factors and forest types. Our findings provide valuable insights into the dynamic nature of macrofungal diversity in the Ancient Tree Park, highlighting the influence of climate and forest type.

## 1. Introduction

Biodiversity, defined as the variety of organisms in all ecosystems, plays a critical role in sustaining important ecological functions such as climate regulation, food production, water and soil conservation, and waste treatment [1]. It is an essential component of natural ecological balance and promotes the harmonious development of humans and nature, making it crucial for biodiversity protection and ecological balance maintenance. Moreover, it is vital for achieving the sustainable development of human society in the future. Despite its significance, global biodiversity faces unprecedented threats due to environmental pollution caused by overpopulation and overdevelopment, man-made genetic contamination, the invasion of alien species, habitat destruction resulting from human activities, global climate change, and geological activity. Human activities have a profound impact on biodiversity and pose a significant threat to species’ survival worldwide [2].

Macrofungi are highly diverse organisms with a wide distribution across various environmental living substrates. These fungi play a crucial role in the cycling of carbon [3], nitrogen [4], and other elements [5,6]. In addition, macrofungi can enhance the resistance of host plants against external harmful substances, offering a defense mechanism against various stressors [7]. Additionally, these fungi play a crucial role in improving soil conditions by actively participating in nutrient cycling, organic matter decomposition, and soil structure enhancement [8]. Through their activities, macrofungi contribute to the overall fertility and health of soils. Moreover, their presence and interactions within ecosystems contribute to the stability and restoration of these systems, making macrofungi key players in maintaining ecological balance and functioning [9]. Saprotrophic macrofungi are also involved in the material cycle and energy flow, such as decomposing fallen timber and dead wood into other substances, such as lignin [10], cellulose [11], and hemicellulose [12], finally converted into glucose, fructose, etc. Thus, studying and protecting macrofungal diversity can help maintain ecosystem balance. Recent studies have identified several factors affecting the diversity of macrofungi, including plant community composition [13,14,15,16], soil physicochemical properties [17,18,19], climatic conditions [20,21,22], organic matter content [23,24], human disturbance [25,26,27], and altitude [28,29]. These findings suggest that a comprehensive understanding of macrofungal diversity requires consideration of multiple factors and highlights the need for ongoing research and conservation efforts.

Soil fungi are critical in regulating ecosystem functions in terrestrial environments, and changes in forest succession can alter aboveground vegetation species composition and soil properties, leading to changes in soil microbial community function [30]. The use of fungicides such as Oxathiapiprolin has been found to have negative impacts on soil fungal communities that play important roles in ecosystem functioning and nutrient cycling. For example, the fungicide was found to decrease the abundance of certain fungal taxa, such as Ascomycota and Basidiomycota, while increasing the abundance of others, such as Mortierellomycota [31]. Similarly, the addition of nitrogen to permafrost peatland soils has been shown to increase the abundance of microbial functional genes associated with bacteria, fungi, archaea, nifH, b-amoA, and mcrA, suggesting that nitrogen addition can stimulate the decomposition of soil organic matter [32]. Meanwhile, the distribution and abundance of ectomycorrhizal and saprotrophic fungi are influenced by elevation and changes in abiotic factors such as temperature, moisture, and soil properties, including pH, soil organic matter, and soil nitrogen content. Although studies have highlighted the ecological roles of Ectomycorrhizal (EcM) and Saprotrophic (SAP) fungi in boreal forests, there is still a lack of research on the impact of soil fungi on macrofungal diversity [33]. As soil fungi play crucial roles in nutrient cycling and ecosystem processes, it is important to further investigate the relationships between soil fungi and macrofungal diversity, especially in the context of changing environmental conditions such as climate change.

Located at an altitude of 822 m in the Changbai Mountain region of northeast China, the Ancient Tree Park is a mixed coniferous forest belt covering a large area of over 11,000 *Pinus koraiensis* trees, as well as other species such as *Quercus mongolica* and *Acer mono*. The region is known for its unique climatic conditions, with abundant precipitation and low temperatures, making it an ideal location for studying fungal species diversity. As the forest succession progresses, changes in aboveground vegetation species composition and soil properties can have a significant impact on the soil microbial community function.

The objectives of this study were as follows: (1) to determine and compare the macrofungi diversity above-ground and the habitat soil fungi community composition belowground across different forest types; (2) to quantitatively assess the relative importance of climate in shaping macrofungal diversity; and (3) to quantitatively assess the relative importance of species correlations of macrofungi with habitat soil fungi to the community structure. The results could establish a baseline for macrofungal diversity and composition in the mature continental mixed coniferous forests of northeastern China.

## 2. Materials and Methods

### 2.1. Study Site and Sporocarp Sampling

The investigation was conducted in mixed forests located in the northeast region of China, specifically in Jilin province. The Changbai Mountain Ancient Tree Park encompasses a diverse range of vegetation types, including broad-leaved forest (B), coniferous forest (C), and mixed forest (M). The B site (127°52′3548″ E, 42°29′16.7244″ N, 821 m) consists of *Quercus mongolica*, *Acer pictum*, *Ulmus laciniata*, and other species of broad-leaved trees. The C site (127°51′38.5812″ E, 42°29′11.4881″ N, 968 m) is characterized by *Pinus koraiensis*, while the M site (127°51′30.1608″ E, 42°29′2.5692″ N, 894 m) is composed of *Quercus mongolica*, *Acer pictum*, *Ulmus laciniata*, *Pinus koraiensis*, and other woody plants (Figure 1). The study sites were not confined to specific, designated plots; instead, the investigation encompassed the entire area of the broad-leaved forest, coniferous forest, and mixed forest within the Changbai Mountain Ancient Tree Park. The data collection process involved random sampling across different areas within each forest type, aiming to obtain a representative sample of macrofungi present in the mixed forests and the various forest types. This approach was chosen to avoid potential bias and to ensure a comprehensive understanding of the macrofungal community composition across the entire study area.

Macrofungi were collected from various sources such as rotten wood, litter, and the ground. The collection was conducted during the months of July to October over a three-year period from 2019 to 2021. During the collection process, photographs were taken to document the habitat vegetation, growing substrate, and morphological characteristics of the collected specimens. To ensure the long-term preservation of the specimens, fresh samples were dried and subsequently preserved in the Herbarium of Jilin Agricultural University (HMJAU), located in Changchun, China.

### 2.2. Species Identification and Community Composition Analysis

Macrofungal specimens, such as fruiting bodies or sporocarps, were collected from the study sites and carefully examined. Species identification was conducted using a combination of ITS barcoding and morphological characteristics, employing taxonomic keys and mycological expertise. The classification of Basidiomycota followed He’s classification system [34], while the classification of Ascomycota was based on Wijayawardene’s system [35]. Each identified species was recorded along with its abundance, providing valuable data for subsequent community composition analysis and insight into the distribution and relative abundance of different macrofungal species in the study area.

DNA was then extracted from the fruiting bodies of the collected specimens using a Hi-DNAsecure Plant kit (TIANGEN, Beijing, China) following the manufacturer’s instructions. The internal transcribed spacer (nrITS) of the rDNA region was amplified using the ITS4 and ITS5 primers [36]. PCR was performed with an ABI 2720 Thermal Cycler (Gene Co. Ltd., Foster City, CA, USA) using a 25 µL reaction system consisting of 2.5 µL 10 × PCR buffer, 1.5 µL of MgCl_2_ (25 mM), 1.25 µL of each primer (10 µm), 0.5 µL of dNTP (10 mM each), 1.25 µL of template DNA, 0.25 µL of Taq polymerase (5 U/µL), and 16.5 µL of ddH_2_O. PCR conditions were as follows: an initial step of 5 min at 95 °C; 30 cycles of 30 s at 95 °C, 40 s at 55 °C, 50 s at 72 °C; followed by 10 min at 72 °C. The PCR products were sent to Sangon Biotech (Shanghai, China) for directional sequencing with the same primers used for the PCR. Consensus sequences were derived for each amplicon using BioEdit 7.2. These consensus sequences were then compared to the NCBI database using BLAST to determine their identity and similarity to known sequences.

We analyzed the community composition of macrofungi using three alpha diversity indices. The Menhinick richness index (R) reflects the species richness of the community, while the Shannon index (D) indicates the diversity of the community species. Pielou’s evenness index (E) reflects the distribution of the number of individuals in each species. The formulae for these diversity indices were as follows: (1)$$R=S/\sqrt{N}$$
(2)$$D=-\sum P_{i}\ln\left(P_{i}\right)\;$$
(3)$$E=H^{\prime }/\ln S;H^{\prime }=-\Sigma P_{i}\ln\left(P_{i}\right)$$
where P_i_ is the proportion of species i to the total number of individuals of all species; ln is the natural logarithm; S is the total number of species; and N is the total number of individuals observed.

Principal Component Analysis (PCA) was performed using OmicStudio [37] to explore variations in macrofungal community composition across three forest types. The data were standardized through z-score normalization to remove scale and magnitude differences and processed to find principal components explaining the variance.

To study the effects of climatic factors on macrofungi composition, the local weather conditions of Changbai Mountain Ancient Tree Park during the period from July to October 2019 to 2021 were obtained by querying the WheatA Agrometeorological big data system http://www.wheata.cn/ (accessed on 10 November 2021), which provides comprehensive climate data for the region. Spearman correlation analysis was used to examine the relationships between the absolute abundance of macrofungal sporocarps and the selected abiotic factors (*p*-values, two-tailed; confidence intervals, 95%) using OmicStudio tools.

The functional prediction of macrofungi was performed using the FUNGuild database http://www.stbates.org/guilds/app.php (accessed on 26 March 2022) [38]. Through this analysis, we identified various ecological functions fulfilled by macrofungi, including ectomycorrhizal fungi, soil saprotrophs, wood-decaying fungi, litter saprotrophs, dung saprotrophs, and entomogenous fungi. Appendix A presents a comprehensive reference to the identified genera and their associated ecological functions, and a detailed list of the identified genera and their corresponding ecological functions.

### 2.3. Habitat Soil Sampling, Identification, and Statistical Analysis

Soil samples were collected from three different forest types monthly, from June to October 2021, resulting in a total of 45 samples (15 samples per forest type). The soil sampling was carried out using a five-point sampling method, and this procedure was repeated three times for each forest type. The five-point sampling method involves collecting soil samples from five different locations within each forest type. This approach ensures that the samples are representative of the overall soil characteristics within the specific forest type. At each sampling spot, soil samples were collected from the overlying soil layer at a depth of 15 cm. Approximately 200 g of soil was collected per sample. To maintain the integrity of the samples, each collected soil sample was placed into a zip-lock bag along with a desiccant. The bags were numbered for identification purposes. 

A total of 45 soil samples were collected from the Ancient Tree Park of Changbai Mountain. DNA was extracted from the soil samples using the E.Z.N.A. Soil DNA kit (D5625, Omega Bio-tek, Inc., Norcross, GA, USA) according to the manufacturer’s instructions, with duplicate extractions performed for each sample. The extracted DNA was used to prepare amplicon libraries targeting the ITS2 region. Gene-specific amplification was performed in triplicate using the ITS1FI2 (5’-GTGARTCATCGAATCTTTG-3’) and ITS2 (5’-TCCTCCGCTTATTGATATGC-3’) [39] primers and the Phusion^®^ Hot Start Flex 2X Master Mix (New England Biolabs, Ipswich, MA, USA). PCR amplification was performed in a total volume of 25 µL of reaction mixture containing 25 ng of template DNA, 12.5 µL of PCR Premix, 2.5 µL of each primer, and PCR-grade water to adjust the volume. The cycling conditions involved an initial denaturation step at 98 °C for 30 s, followed by 35 PCR cycles consisting of 98 °C for 10 s, 54 °C for 30 s, and 72 °C for 45 s, with a final extension at 72 °C for 10 min. Amplification products were submitted to the Illumina NovaSeq PE250 platform for high-throughput sequencing (LC-BIO, Hangzhou, China). Quantitative Insights Into Microbial Ecology 2 (QIIME2) was used to analyze the sequence data [40].

Raw sequence data from the Novaseq platform were initially managed by cutadapt software (version 1.9) to remove the barcodes [41]. Then, PEAR software (v0.9.6) was used to merge the pair-end reads with a default error matching rate of 0.25 [42]. Low-quality paired-end reads were removed by fqtrim software (version 0.94) with the parameters “-P 33 -w 100 -q 20 -l 100 -m 5 -p 1 -V -o trim.fastq”. Chimeras were deleted using Vsearch software (version 2.3.4) with default parameters [43]. The Denoising Algorithm (DADA2) was employed for clustering based on representative sequences with single-base accuracy. The process involved using the “Dereplication” approach, which is equivalent to clustering with 100% similarity. As a result, Amplicon Sequence Variants (ASVs) were generated, representing unique sequence variants within the dataset. Taxonomy was assigned to ASVs using Ribosomal Database Project RDP Classifier trained on the database UNITE using 0.7 confidence values. The obtained ASVs abundance table was used to generate a visual representation of the number of ASVs common and unique to each group using a Venn diagram. In the diagram, each circle represents a group, and the number of circles and overlapping parts indicates the number of ASVs shared by the respective samples. The parts without overlaps represent the number of ASVs unique to each group.

Fungal alpha diversity was assessed using the Chao1 index, Shannon index, Simpson index, and Pielou’s evenness index. The Chao1 index provided an estimate of species richness, while the Shannon index measured both richness and evenness. The Simpson index indicated community diversity, with higher values indicating greater diversity. Pielou’s evenness index evaluated the distribution of individuals among species. 

### 2.4. Correlations between Above-Ground Macrofungi and Below-Ground Habitat Soil Fungi

A Spearman correlation analysis was performed to explore the relationship between the relative abundance of macrofungi and habitat soil fungi. This statistical analysis enabled the identification of potential correlations and ecological interactions between macrofungi and habitat soil fungi. By assessing the strength and direction of the correlation, we gained insights into the potential dependencies and associations between above-ground macrofungi and below-ground habitat soil fungi in the studied ecosystem. Analysis was performed using the OmicStudio tools [37].

## 3. Results

### 3.1. Macrofungal Species Composition in Ancient Tree Park

Over a three-year period from 2019 to 2021, we collected a total of 1007 macrofungi specimens from various habitats, including broad-leaved forests, coniferous forests, and mixed forests. Through the combined analysis of morphological characteristics and ITS sequencing, we successfully identified and classified 210 distinct fungal species within Changbai Mountain Ancient Tree Park. The species represented 2 phyla, 6 classes, 18 orders, 55 families, and 94 genera (Table 1). The fungal diversity analysis revealed distinctive genera within each forest type. In the mixed forest, a wide range of unique genera were identified, including *Arrhenia*, *Conocybe*, *Entocybe*, *Ampulloclitocybe*, *Hypsizygus*, *Lyophyllum*, *Rhodocollybia*, *Hohenbuehelia*, *Schizophyllum*, *Gymnopilus*, *Notholepista*, *Singerocybe*, *Auricularia*, *Boletus*, *Geastrum*, *Porodaedalea*, *Rigidoporus*, and *Postia*. These genera demonstrate the richness and uniqueness of fungal species thriving in the mixed forest environment. The conifer forest exhibited its own set of distinct genera, including *Bulgaria*, *Ophiocordyceps*, *Pseudosperma*, *Calocybe*, *Panellus*, *Lacrymaria*, *Pterula*, *Pseudoomphalina*, *Clavulina*, and *Thelephora*. These unique genera underscore the specific fungal communities associated with coniferous habitats. In contrast, the broad-leaved forest showed a comparatively lower number of unique genera, with *Crinipellis*, *Volvopluteus*, *Harmajaea*, *Stropharia*, *Tubaria*, *Phallus*, *Laetiporus*, and *Bjerkandera* being the eight identified. Although fewer in number, these unique genera contribute to the overall fungal diversity within the broad-leaved forest ecosystem. Some representative species are shown in Figure 2.

Among the identified macrofungi species, there were four dominant families that stood out with a number of species equal to or exceeding 15. The richness percentages and rankings of these families are shown in Table 2. Russulaceae exhibited the highest richness, accounting for 9.1% of the total identified species. Following closely behind was Tricholomataceae, representing 8.6% of the identified species. Mycenaceae and Strophariaceae showed an equal percentage, each contributing to 7.6% of the identified species. In contrast, 51 other families, each with less than 15 species, collectively accounted for 67.1% of the identified species.

In terms of dominant genera, three stood out with a significant presence, each having 10 or more species. These included *Mycena*, *Clitocybe*, and *Russula*. These genera showed the diversity and abundance of macrofungi within the study area. Additionally, 29 genera contained 2–4 species, representing 30.85% of the total genera and 37.62% of the identified species. Furthermore, a considerable number of genera, 55 in total, were represented by only one species, making up 57.89% of the genera but contributing to 26.19% of the identified species.

### 3.2. Dynamic Change in Macrofungal Composition in the Ancient Tree Park

#### 3.2.1. Macrofungal Composition in Different Vegetation Types

To investigate the impact of different vegetation types on macrofungal community composition, we conducted a comparison among the forest types. The alpha diversity analysis, as depicted in Figure 3a, revealed that the diversity of the broad-leaved (B) and mixed (M) forests exhibited similarities, while both differed from the coniferous (C) forest, although the differences were not statistically significant (*p* < 0.05). Regarding species richness, the order of increase was C < B < M. The mixed forest (M) displayed the highest richness indices, with 7.30 and 4.61, accounting for 69.52% of the total species. In comparison, the broad-leaved forest (B) contained 104 species, representing 49.52% of the total species. Vegetation type C comprised 101 species with richness indices of 5.98 and 3.71, constituting 48.10% of the total species. These findings suggest macrofungi’s preference for mixed forests in the study area. Furthermore, the evenness index and macrofungal diversity, as indicated by the richness (R) and diversity (D) indices, were highest in the mixed forest (M).

The PCA results shown in Figure 3b demonstrate some similarities in macrofungal community composition among the broad-leaved (B), coniferous (C), and mixed (M) forests, indicating a degree of similarity in macrofungal community compositions corresponding to different elevational distributions. However, the macrofungal community composition of the broad-leaved forest (B) is notably distinct from that of the coniferous forest (C), indicating a significant difference between these two forest types. 

Based on our findings, the coniferous forest (C) exhibited lower macrofungal species richness compared to the other vegetation types, while the mixed forest (M) displayed the highest evenness and greatest macrofungal diversity, as indicated by the Richness index and Shannon index.

Regarding the species compositions of the three forest types, we observed a consistent trend in which the compositions of the broad-leaved forest (B) and coniferous forest (C) exhibited similarities but differed from the mixed forest (M), as shown in Figure 4a. In the broad-leaved forest (B), the genera *Mycena* (10 species) and *Cortinarius* (6 species) were the richest, with the number of species exceeding five. Similarly, in the coniferous forest (C), the genera *Mycena* (10 species), *Russula* (8 species), and *Cortinarius* (6 species) were the richest. In the mixed forest (M), the genera *Russula* (12 species), *Mycena* (9 species), *Clitocybe* (9 species), *Armillaria* (5 species), and *Marasmius* (5 species) exhibited the highest richness.

Across all three forest types, the main functional groups of macrofungi included ectomycorrhizal fungi, wood-decaying fungi, soil saprotrophic fungi, litter saprotrophic fungi, and entomogenous fungi. The number of wood-decaying fungi species was the highest, with a total of 121 species, and this number increased in the order of coniferous forest (C) < broad-leaved forest (B) < mixed forest (M) (Figure 4b). In the mixed forest (M), the number of ectomycorrhizal fungi, wood-decaying fungi, soil saprotrophic fungi, and litter saprotrophic fungi was higher compared to the other forest types. In the coniferous forest (C), wood-decaying fungi and litter saprotrophic fungi showed higher species numbers. The genera *Russula* and *Cortinarius* exhibited the highest occurrence among ectomycorrhizal fungi. *Armillaria*, *Xylaria*, and *Pleurotus* were the most common wood-decaying fungi. *Mycena*, *Marasmius*, and *Clitocybe* were the dominant litter saprotrophic fungi.

#### 3.2.2. The Effects of Climatic Factors on the Macrofungi Composition

The results of the relationships among climates (rainfall and temperature), species richness, and the number of specimens are presented in Figure 5. Figure 5a demonstrates a positive correlation between mean monthly rainfall and both the number of species and absolute abundance. This suggests that higher levels of rainfall are associated with increased species richness and abundance. Similarly, Figure 5b shows that mean monthly temperature also has a positive effect on the number of species and absolute abundance. However, the correlation with temperature is relatively weaker compared to the correlation with rainfall.

Through Spearman correlation analysis (Figure 5c), we identified several genera that exhibited positive correlations with mean monthly temperature: Hericium, Hohenbuehelia, Lacrymaria, Pterula, Crepidotus, Psathyrella, Gymnopus, Dacryopinax, Laccaria, Phallus, Elmerina, and Ramaria. On the other hand, genera like Hypsizygus, Pholiota, Cystoderma, Tricholoma, Schizophyllum, Auricularia, and Amanita showed negative correlations with mean monthly temperature.

Additionally, as shown in Figure 5d, the absolute abundance of *Pterula*, *Agaricus*, *Infundibulicybe*, *Clitocybe*, *Lycoperdon*, *Melanoleuca*, *Pluteus*, *Lepiota*, *Clavulina*, and *Callistosporium* displayed positive correlations with mean monthly rainfall. Conversely, genera such as *Xeromphalina*, *Hypsizygus*, *Tricholoma*, *Cystoderma*, *Schizophyllum*, *Auricularia*, *Sparassis*, *Daldinia*, and *Phallus* showed negative correlations with mean monthly rainfall.

### 3.3. Soil Fungal Community Structure in Habitats

A total of 3,661,298 raw reads were obtained from the sequencing analysis, with an average of 81,362 effective sequences per sample. The effective rates of the data were all above 95%, indicating high-quality sequencing data. Additionally, the Q20 value, which represents the percentage of bases with a quality score of 20 or higher, was greater than 95.85%, and the Q30 value, which represents the percentage of bases with a quality score of 30 or higher, was greater than 86.67%. Regarding the identification of fungi communities, a total of 11,294 Amplicon Sequence Variants (ASVs) were obtained (Figure 6a). Among them, 3433 ASVs were derived from forest type B, 2909 ASVs from forest type C, and 3851 ASVs from forest type M. Notably, 1080 ASVs were found to be shared across all three forest types, indicating a degree of similarity in fungal composition and potential ecological interactions.

We conducted an alpha diversity analysis on 45 soil samples based on the results of the ASV clustering analysis, as described in Figure 6b. The observed ASV count, as measured by the Chao1 index, followed the order of C < B < M, indicating that the mixed forest (M) had the highest number of species. Furthermore, the Shannon diversity index, which accounts for both species richness and evenness, was highest in the mixed forest (M) compared to the other two vegetation types. This suggests that the mixed forest exhibited the highest level of species diversity among the three forest types. Similarly, the Simpson index, which quantifies the dominance or evenness of species in a community, was also highest in the mixed forest (M). These findings indicated that the mixed forest harbored a greater variety of species and a more balanced distribution of individuals compared to the broad-leaved forest (B) and coniferous forest (C).

A total of 609 fungal genera were identified (Figure 6c), with *Russula*, *Inocybe*, *Sebacina*, *Cryptococcus*, and *Tomentella* being the top five genera in terms of relative abundance. From the perspective of forest type, Bray–Curtis distance analysis revealed that the soil fungi within the vegetation types were predominantly divided into two categories. Specifically, broad-leaved forest and mixed forest were clustered in the same branch, indicating that the fungal communities of the two forest types had the highest degree of similarity, while coniferous forest was separated as an individual branch, indicating significant differences in species composition from the other two forest types. The results revealed that *Russula*, *Tricholoma*, *Amanita*, and *Inocybe* were the dominant genera in the coniferous forest. Meanwhile, *Scutellinia*, *Hymenogaster*, *Sebacina*, and *Thelephora* were the dominant genera in broad-leaved forests. Similarly, *Cryptococcus*, *Agrocybe*, *Subulicystidium*, *Humaria*, *Otidea*, and *Haptocillium* were the dominant genera in the mixed forest.

### 3.4. Relationships of Above-Ground Macrofungi with Soil Fungi in the Habitat 

To examine the relationship between above-ground macrofungi and soil fungi at the genus level, a correlation heatmap analysis was conducted (Figure 7). The purpose was to identify differences in the associations between these two groups of fungi. The results of the correlation analysis revealed interesting findings, including significant correlations between several genera. For example, *Agrocybe*, *Amanita*, *Entoloma*, *Hypholoma*, *Inocybe*, *Lentinellus*, *Lycoperdon*, *Mucidula*, *Mycena*, *Peniophora*, *Pluteus*, *Ramaria*, *Sparassis*, *Stropharia*, *Tephrocybe*, and *Tricholoma* exhibited positive correlations between macrofungi and soil fungi. On the other hand, some genera, like *Crepidotus*, *Ganoderma*, *Gymnopus*, *Ophiocordyceps*, *Otidea*, *Pholiota*, *Pleurotus*, *Psathyrella*, *Russula*, *Trametes*, *Xeromphalina*, and *Xylaria* showed negative correlations between macrofungi and soil fungi. Detailed correlation coefficients (rho) and additional information are shown in Appendix A.

## 4. Discussion

### 4.1. The Community Composition of Macrofungi Varied with Forest Types

This study conducted a comprehensive analysis of macrofungal diversity in the Changbai Mountain Ancient Tree Park, located in Jilin Province, China. The main objective was to evaluate the correlations between macrofungal diversity and forest types by comparing species richness and composition structure. The forests were categorized into three types: broad-leaved forests (B), coniferous forests (C), and mixed forests (M), enabling an analysis of macrofungal composition across different forest types. The results revealed that the species richness was highest in the mixed forests (M), with a recorded count of 146 species. This finding aligns with previous studies that have reported higher macrofungal diversity in mixed forests and identified the presence of both widely distributed and narrowly distributed macrofungal types [44], corroborating our findings. The composition of macrofungal species varied among the three forest types. Only 13 macrofungal species were found to be shared between the broad-leaved and coniferous forest types. This supports previous studies that have reported variation in the distribution of macrofungal species across different habitat types and functional groups [45], which is consistent with our findings.

To explore the differences in macrofungal species composition among the three forest types, we investigated their ecological functions. Our study found that ectomycorrhizal fungi, wood-decaying fungi, soil saprotrophs, and litter saprotrophs were predominantly present in mixed forests (Figure 4b). Ectomycorrhizal fungi, such as *Lactarius* [46,47], *Amanita* [48,49], *Russula* [50,51], and *Cortinarius* [52,53], are known to associate primarily with *Quercus* and *Pinus* trees. Since mixed forests in our study area consist of *Quercus mongolica* and *Pinus koraiensis*, it is likely that the presence of these tree species contributes to the higher abundance of ectomycorrhizal fungal species. Wood-decaying fungi played a significant role, with 75 species observed, mainly distributed in mixed forests (51 species) and broad-leaved forests (40 species). Wood-decaying fungi tend to exhibit preferences for specific vegetation types under similar climatic conditions, often growing on larger fallen wood. For instance, species like *Hypholoma capnoides*, *Xylaria longipes*, *Psathyrella candolleana*, and *Hypholoma fasciculare* were mainly found on *Quercus* woods in our study. Other studies have also highlighted the close relationship between fungi and dominant tree species, as enzymes produced by fungi have adapted to wood with distinct chemical and physical properties [54]. Additionally, the requirement for space may contribute to the prevalence of species that produce large fruiting bodies [55].

### 4.2. Climate Affects the Composition of Macrofungi

Our study focused on the Ancient Tree Park in Jilin Province, China, to explore the relationship between macrofungi and climatic conditions [56,57,58,59]. By collecting data on macrofungi species and their abundance on a monthly basis, we conducted a correlation analysis with average rainfall and temperature (Figure 5). The results revealed a positive correlation between mushroom species richness and both monthly average rainfall and temperature. This finding aligns with previous studies conducted by Angelini et al. [60] and Zhang et al. [61], who highlighted the significant impact of meteorological conditions, particularly rainfall levels and relative humidity, on fungal community structure. Additionally, our study found a positive correlation between monthly mean temperature and overall macrofungal abundance during the growing season (June–October). It is worth noting that while some studies have reported a negative correlation between macrofungal species richness and temperature [62], the contrasting findings may be attributed to variations in the geographical location, research scale, and seasonal span of the study sites [63,64]. Another possible explanation is the time required for spores to germinate and produce fruiting bodies [65], leading to a lag in the growth of macrofungal abundance in response to temperature changes [56]. The variability in species diversity among six sites examined by Hu et al. [66] further supports the positive correlation between temperature and macrofungal growth. Overall, our results are consistent with the findings of Hu et al. [66], indicating the significance of temperature in influencing macrofungal dynamics.

### 4.3. Soil Banks Were Not a Major Factor in the Variation in Macrofungal Diversity

It is generally believed that the soil fungal bank is the main source of above-ground macrofungi and the main factor affecting their distribution [67,68]. Our survey findings indicate that the diversity of soil fungi within the same conservation area remained relatively consistent across different forest types, with no significant statistical differences observed. Therefore, soil banks were not identified as a major factor contributing to the variation in macrofungal diversity at the protected area scale, which does not contradict other ideas that suggest that changes in macrofungal community structure are less influenced by soil factors at small scales where soil factors are consistent. Interestingly, our study also revealed significant differences in the community structure of fungi between above-ground and below-ground environments, highlighting the distinct ecological dynamics and potential interactions between these two fungal communities. These results emphasize the importance of considering both above-ground and below-ground fungal communities when assessing and managing macrofungal diversity in protected areas.

Our study revealed notable differences between macrofungi and the top three genera with the highest relative abundance of soil fungi under the same external conditions. In 2021, the genera *Ramaria*, *Mycena*, and *Xylaria* exhibited the highest relative abundance among macrofungi, whereas *Tricholoma*, *Inocybe*, and *Russula* were the dominant genera among soil fungi. These findings suggest that soil fungi may not be the sole source of above-ground macrofungi. Research has indicated that macrofungi aboveground can disperse their spores over long distances through wind, leading to species migration. This phenomenon may contribute to the observed differences between above-ground and soil fungal communities [69,70,71].

In soil microecology, the interrelationships between species are more complex. We found that some underground species have a certain impact on the diversity of macrofungi, which may reflect the competition or synergistic relationship between species. We found that the relationship between the genera of macrofungi and the related genera of soil fungi is not unique, indicating that the relationship between the same groups in different ecological niches is not necessarily positive. In our results, the *Russula* of macrofungi was negatively correlated with the *Russula* in soil, indicating that there may be competition between macrofungi and the same genus of soil fungi. *Russula*, as a typical ECM fungus, has a high relative abundance (49.27%) in soil, but a low relative abundance (2.91%) in large fungal communities. The reason for this phenomenon may be that ECM in the soil is affected by various factors during the formation of fruiting bodies. In addition, we found a significant positive correlation between the *Russula* of macrofungi and several genera (*Tricholoma*, *Pluteus*, *Sparassis*, *Agrocybe*, and *Lycoperdon*) in soil (Sig. = 0.04). These results suggest that macrofungi may interact with many taxa in the soil during their growth. Previous studies have shown the co-occurrence of *Cortinarius* with *Russula* in root systems across various forest types, highlighting the complex relationships between these genera [72,73]. It is important to note that our results are based solely on statistical analysis, and further investigations are needed to understand the nature of these relationships and the specific manner in which these genera affect *Russula* fungi.

Additionally, our study uncovered intriguing correlations between specific above-ground fungal genera and soil fungal genera, indicating potential mutual promotion or inhibition between them. For instance, we observed a significant positive correlation between the above-ground genus *Xeromphalina* and the soil genus *Tricholoma*, suggesting a potential promotion between these two genera. Similarly, *Russula* above-ground exhibited a positive correlation with *Pluteus* from the soil, and *Tephrocybe* above-ground showed a positive correlation with *Agrocybe* from the soil, among other notable associations. These findings suggest that specific fungal genera in the soil may play a role in promoting the growth and abundance of particular above-ground genera. Conversely, the study also identified significant negative correlations between certain above-ground genera and soil genera. For example, *Crepidotus* above-ground displayed a significant negative correlation with *Entoloma* from the soil, *Russula* above-ground correlated negatively with *Inocybe* from the soil, and *Pluteus* above-ground exhibited a negative correlation with *Crepidotus* from the soil, among others. These results suggest that certain fungal genera in the soil may inhibit the growth or abundance of specific above-ground genera.

The results highlight the remarkable fungal diversity present in Changbai Mountain Ancient Tree Park and emphasize the influence of forest type in shaping fungal communities. These findings contribute to our understanding of forest ecology and have implications for conservation and management efforts in the region.

## 5. Conclusions

The occurrence of macrofungi is closely related to vegetation. By comparing three different forest types, we found that the mixed forest had the highest species richness of macrofungi. Moreover, the most abundant ecological function groups of macrofungi were wood-decaying fungi, litter saprotrophic fungi, and ectomycorrhizal fungi. Rainfall and temperature are important factors affecting the composition and overall abundance of macrofungal communities during the growing season, but the specific effects vary with different species. At the same time and in the same place, the community composition of macrofungi and soil fungi is not exactly the same, and the correlation between the same genus of macrofungi located in two different ecological niches is not necessarily positive, and there may be cooperative or competitive relationships among multiple genera.

## Figures and Tables

**Figure 1 jof-09-00856-f001:**
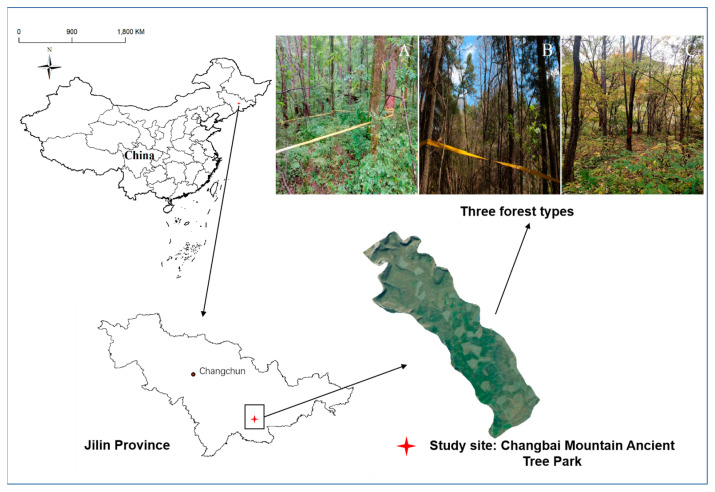
Geographical location of the Changbai Mountain Ancient Tree Park in relation to the China map, the Jilin Province map, and the three forest types we studied. (**A**) Broad-leaved forest; (**B**) coniferous forest; (**C**) mixed forest.

**Figure 2 jof-09-00856-f002:**
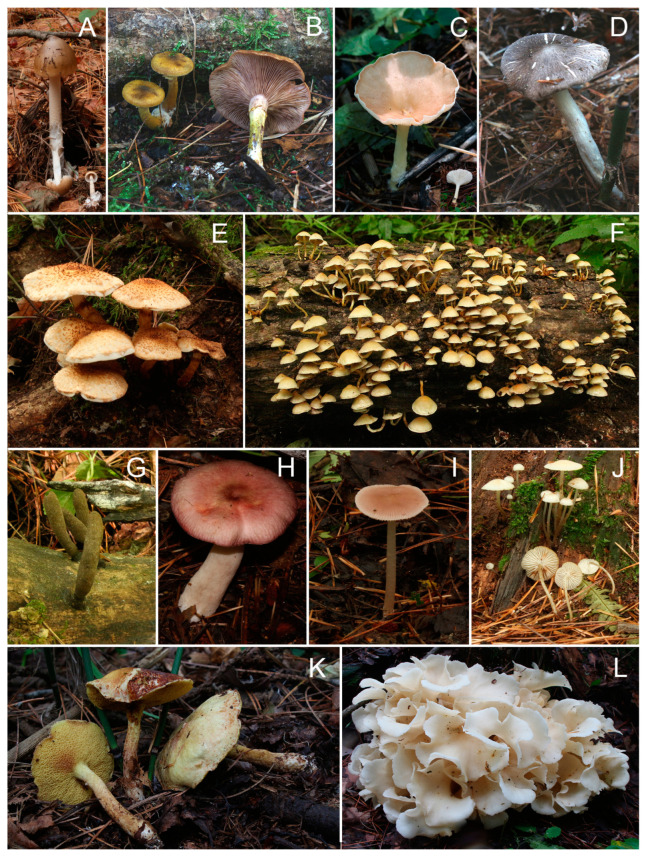
A collection of photographs showing various macrofungi species found in the Ancient Tree Park. (**A**) *Amanita fulva*; (**B**) *Armillaria gallica*; (**C**) *Clitocybe gibba*; (**D**) *Tricholoma terreum*; (**E**) *Pholiota squarrosa*; (**F**) *Hypholoma fasciculare*; (**G**) *Xylaria longipes*; (**H**) *Russula madrensis*; (**I**) *Mycena pura*; (**J**) *M. laevigata*; (**K**) *Boletus kauffmanii*; (**L**) *Sparassis latifolia*.

**Figure 3 jof-09-00856-f003:**
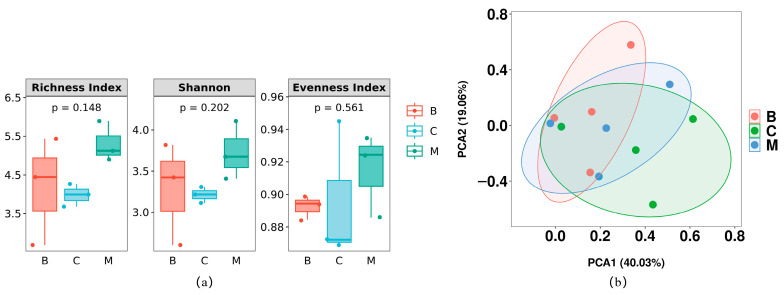
Alpha diversity analysis (**a**) and PCA analysis (**b**) of macrofungi in three forest types. B: Broad-leaved forest; C: coniferous forest; M: mixed forest. Significance was determined by the Kruskal–Wallis test with Dunn’s multiple comparison test. (*p* < 0.05). The sample sizes are as follows: B (*n* = 3), C (*n* = 3), M (*n* = 3).

**Figure 4 jof-09-00856-f004:**
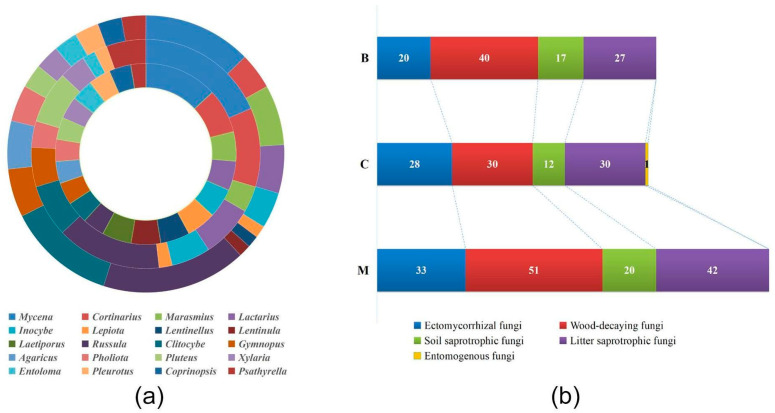
The circular map of macrofungal species (**a**) and ecological function groups (**b**) in different forest types. The outer ring (**a**) represents the species composition in the mixed forest, the middle ring (**a**) represents the species composition in the conifer forest, and the inner ring (**a**) represents the species composition in the broad-leaved forest. The values displayed on the map (**b**) indicate the cumulative number of macrofungal functional groups within each forest type (B: broad-leaved forest; C: coniferous forest; M: mixed forest).

**Figure 5 jof-09-00856-f005:**
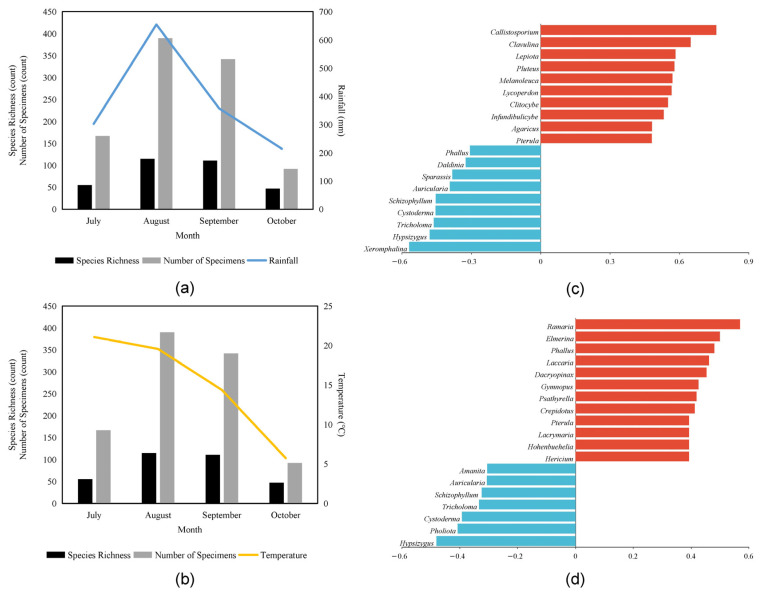
Relationships among climate, species richness, and number of specimens. (**a**) Relationships among species richness, number of specimens, and rainfall. (**b**) Relationships among species richness, number of specimens, and temperature. (**c**) Spearman correlation analysis of macrofungal abundance and rainfall. (**d**) Spearman correlation analysis of macrofungal abundance and temperature. The horizontal axis represents the value of correlation coefficients, and the vertical axis represents the genera of macrofungi.

**Figure 6 jof-09-00856-f006:**
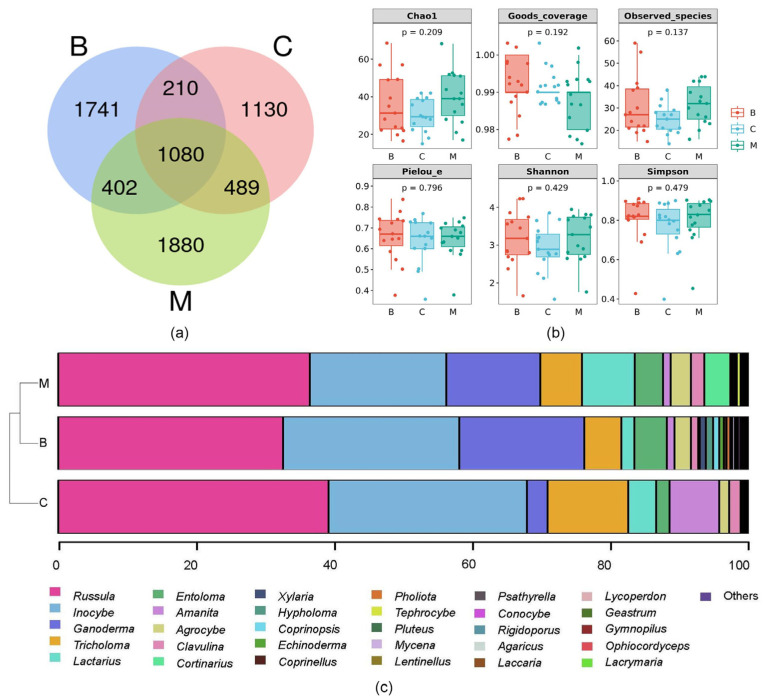
Bioinformatic analysis of soil fungi under different forest types. (**a**) Venn diagram of Amplicon Sequence Variants; (**b**) alpha diversity analysis; (**c**) relative abundance of fungal communities at the genus level. B: broad-leaved forest; C: coniferous forest; M: mixed forest.

**Figure 7 jof-09-00856-f007:**
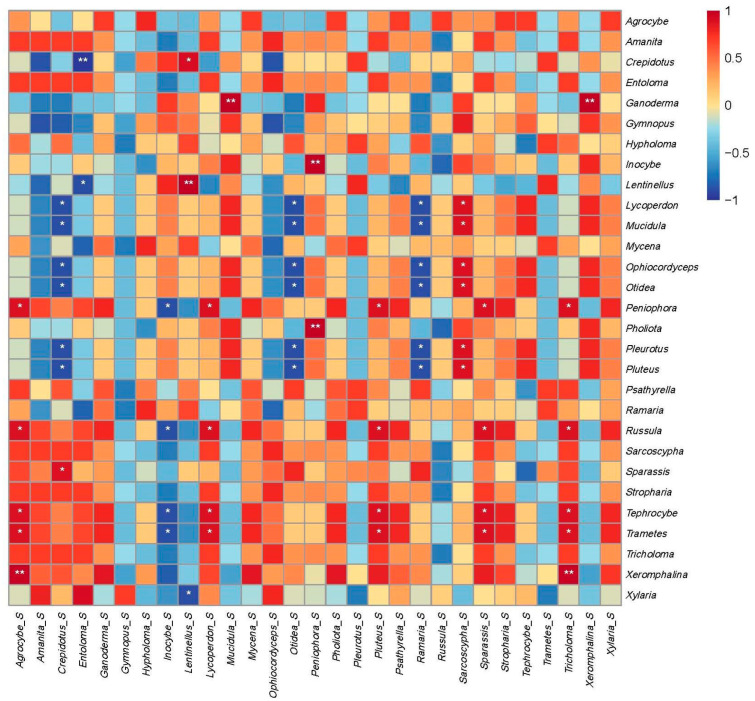
Correlation heatmap of macrofungi and soil fungi. The x-axis of the heat map represents different soil fungal genera, while the y-axis represents macrofungal genera. Each cell in the heat map represents the correlation coefficient between a specific pair of macrofungi and soil fungi genera. The cells are color-coded to indicate the strength and direction of the correlation. A dark shade of red indicates a strong positive correlation, a dark shade of blue represents a strong negative correlation, and neutral colors signify no correlation. A single star (*) indicates that the correlation between the corresponding soil fungal genus and the macrofungal genus is statistically significant at the *p* < 0.05 level. Two stars (**) indicate that the correlation is even more statistically significant at the *p* < 0.005 level.

**Table 1 jof-09-00856-t001:** Macrofungal community compositions in different vegetation types. B: Broad-leaved forest; C: coniferous forest; M: mixed forest.

Phylum	Class	Order	Family	Genus	Vegetation Types
Ascomycota	Leotiomycetes	Helotiales	Bulgariaceae	*Bulgaria*	C
	Pezizomycetes	Pezizales	Pyronemataceae	*Otidea*	C, M
			Sarcoscyphaceae	*Sarcoscypha*	B, M
	Sordariomycetes	Hypocreales	Ophiocordycipitaceae	*Ophiocordyceps*	C
		Xylariales	Hypoxylaceae	*Daldinia*	B, M
			Xylariaceae	*Xylaria*	B, C, M
Basidiomycota	Agaricomycetes	Agaricales	Agaricaceae	*Agaricus*	B, M
				*Echinoderma*	B, M
				*Lepiota*	B, C, M
			Amanitaceae	*Amanita*	B, C, M
			Arrhenieae	*Arrhenia*	M
			Bolbitiaceae	*Conocybe*	M
			Callistosporiaceae	*Callistosporium*	B, C
			Cortinariaceae	*Cortinarius*	B, C, M
			Crepidotaceae	*Crepidotus*	B, C, M
			Entolomataceae	*Entocybe*	M
				*Entoloma*	B, C, M
			Hydnangiaceae	*Laccaria*	C, M
			Hygrophoraceae	*Ampulloclitocybe*	M
			Inocybaceae	*Inocybe*	B, C, M
				*Pseudosperma*	C
			Lycoperdaceae	*Lycoperdon*	C, M
			Lyophyllaceae	*Calocybe*	C
				*Hypsizygus*	M
				*Lyophyllum*	M
				*Tephrocybe*	C, M
			Marasmiaceae	*Atheniella*	M
				*Campanella*	C, M
				*Crinipellis*	B
				*Marasmius*	B, C, M
			Mycenaceae	*Mycena*	B, C, M
				*Panellus*	C
				*Xeromphalina*	C, M
			Omphalotaceae	*Gymnopus*	B, C, M
				*Lentinula*	B, M
				*Marasmiellus*	C, M
				*Rhodocollybia*	M
			Physalacriaceae	*Armillaria*	B, C, M
				*Oudemansiella*	B, C, M
			Pleurotaceae	*Hohenbuehelia*	M
				*Pleurotus*	B, C, M
			Pluteaceae	*Pluteus*	B, C, M
				*Volvopluteus*	B
			Psathyrellaceae	*Coprinellus*	B
				*Coprinopsis*	B, M
				*Lacrymaria*	C
				*Psathyrella*	B, C, M
			Pseudoclitocybaceae	*Harmajaea*	B
			Pterulaceae	*Pterula*	C
			Schizophyllaceae	*Schizophyllum*	M
			Squamanitaceae	*Cystoderma*	B, C, M
			Strophariaceae	*Agrocybe*	B, M
				*Galerina*	B, M
				*Gymnopilus*	M
				*Hebeloma*	B, C
				*Hypholoma*	B, C, M
				*Pholiota*	B, C, M
				*Stropharia*	B
			Tricholomataceae	*Clitocybe*	B, C, M
				*Infundibulicybe*	C, M
				*Lepista*	C, M
				*Melanoleuca*	B, C, M
				*Notholepista*	M
				*Pseudoomphalina*	C
				*Singerocybe*	M
				*Tricholoma*	B, C
			Tubariaceae	*Tubaria*	B
		Auriculariales	Auriculariaceae	*Auricularia*	M
		Boletales	Boletaceae	*Boletus*	M
		Cantharellales	Hydnaceae	*Clavulina*	C
		Geastrales	Geastraceae	*Geastrum*	M
		Gloeophyllales	Gloeophyllaceae	*Neolentinus*	B, C, M
		Gomphales	Gomphaceae	*Ramaria*	B, C, M
		Hymenochaetales	Hymenochaetaceae	*Onnia*	B, C, M
				*Porodaedalea*	M
		Phallales	Phallaceae	*Phallus*	B
		Polyporales	Ganodermataceae	*Ganoderma*	B, M
			Laetiporaceae	*Laetiporus*	B
			Meripilaceae	*Rigidoporus*	M
			Phanerochaetaceae	*Bjerkandera*	B
			Polyporaceae	*Picipes*	B, C
				*Trametes*	B, C, M
				*Trichaptum*	B, C, M
			Postiaceae	*Postia*	M
			Sparassidaceae	*Sparassis*	B, M
		Russulales	Auriscalpiaceae	*Lentinellus*	B, M
			Hericiaceae	*Hericium*	B, M
			Russulaceae	*Lactarius*	B, C, M
				*Russula*	B, C, M
			Stereaceae	*Stereum*	B, M
		Thelephorales	Thelephoraceae	*Thelephora*	C
	Dacrymycetes	Dacrymycetales	Dacrymycetaceae	*Calocera*	C, M
				*Dacryopinax*	B, M
	Tremellomycetes	Tremellales	Aporpiaceae	*Elmerina*	C, M

**Table 2 jof-09-00856-t002:** Dominant families and genera of macrofungi in the Ancient Tree Park.

Dominant Family	Number of Species	Relative Proportions (%)	Dominant Genus	Number of Species	Relative Proportions (%)
Russulaceae	19	9.05%	*Russula*	14	6.67%
Tricholomataceae	18	8.57%	*Mycena*	12	5.71%
Mycenaceae	16	7.62%	*Clitocybe*	10	4.76%
Strophariaceae	16	7.62%			
Total	69	32.86%	Total	36	17.14%

## Data Availability

Not applicable.

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
