# Peer review of "Forest Type and Climate Outweigh Soil Bank in Shaping Dynamic Changes in Macrofungal Diversity in the Ancient Tree Park of Northeast China"

_jof, 2023, doi:10.3390/jof9080856_

Round 1

Reviewer 1 Report

Dear Authors,

Please find my comments on the attached PDF. In my opinion, significant revisions are needed before this manuscript can be considered for publication. I have several concerns, particularly regarding the HTS data and statistical analyses. For instance, it is unclear how many raw reads were obtained in total and how much data was lost during the merging of paired-end reads. Additionally, the manuscript lacks information on the database used for assigning taxonomy to the ASVs, among other crucial details (please refer to the PDF). When describing a pipeline, it is essential to provide all the necessary information for the sake of reproducibility. Simply citing a software or online resource is insufficient. I hope you will find my comments helpful in revising the manuscript.

Best regards,

The English is okay. My major concern here is the science and not language.

Author Response

Dear Reviewer,

I want to express my sincere gratitude for your valuable and insightful review of my [article/paper/research]. Your expertise and thoughtful feedback have been instrumental in improving the quality and clarity of the manuscript. Your commitment to the peer-review process is greatly appreciated, and I am truly thankful for the time and effort you dedicated to this review. Your contributions have been invaluable, and I am grateful for the opportunity to benefit from your expertise. Please see the attachment. Thank you for helping to enhance the overall quality of my work.

Best regards,

Zhaoxiang

Reviewer 2 Report

This study is very important and relevant. The work organically combines classical methods for studying the biodiversity of macrofungi and modern metagenomic environmental technologies. A large and high-quality statistical processing of the obtained data was carried out. Interrelations of species are revealed and the factors influencing are estimated. The appendix contains an impressive list of identified mushroom species.

The text needs a little improvement. In Materials and Methods Study Site is not fully described, the size of the studied sites is not indicated. It was limited plots or all areas broad-leaved forest, coniferous forest, and mixed forest?

The identification of Macrofungal specimens was carried out at a high methodological level using a combination of ITS barcoding and morphological characteristics. However, neither in the text nor in the appendix are the nucleotide sequence numbers deposited in the GenBank databases.

The listing of names and coefficients in lines 339-358 and 412-423 would probably be better presented in the form of tables.

Author Response

(The authors gave the same response as above.)

Reviewer 3 Report

line 19. Replace "analyzed" by "analyzing". It is not clear what "change rules" means; this sentence needs rewriting.

line 20. Replace "compared" by "comparing".

line 265, Table 2. "proportions" is misspelled in two columns.

lines 340-423. There is extensive use of the symbols Pr and Sig starting with line 340. These are misleading. Pr suggest a probability, but it is actually a correlation coefficient, which can take on negative values. On the other hand, Sig is a probability, a p-value. I suggest that all occurrences of "Pr ="

be replaced by 'r =' and that all occurrences of "Sig. =" be replaced by "p =" or "p-value =" between lines 340 to 423.

lines 443-444. This should read "Detailed correlation coefficients (rho) and additional information are shown in Appendix B."

line 533. It should read "fruit bodies" or "fruit-bodies", not "fruity bodies".

Supplementary file:

        Appendix A. Replace "Genera" by "Genus".

        Appendix B. Replace "Pr" by "r" and replace "Sig" by "p" or "p-value". Reduce the number of decimal places to at most 3 or 4 places throughout the whole of this appendix. In the last column, replace "negtive" by "negative" in all occurrences.

Author Response

(The authors gave the same response as above.)

Round 2

Reviewer 1 Report

Thanks to the authors for kindly adressing my suggestions on the previous version of the manuscript. I am happy with the changes made. I don't have any further comments or suggestions. Best wishes!

Minor language editing can be done during the proof reading.